# Modeling Car-Following Behaviors and Driving Styles with Generative Adversarial Imitation Learning

**DOI:** 10.3390/s20185034

**Published:** 2020-09-04

**Authors:** Yang Zhou, Rui Fu, Chang Wang, Ruibin Zhang

**Affiliations:** 1School of Automobile, Chang’an University, Xi’an 710064, China; 2016022005@chd.edu.cn (Y.Z.); wangchang@chd.edu.cn (C.W.); 2016022016@chd.edu.cn (R.Z.); 2School of Vehicle Engineering, Xi’an Aeronautical University, Xi’an 710077, China

**Keywords:** human-like car-following model, driving styles, generative adversarial imitation learning, gated recurrent units

## Abstract

Building a human-like car-following model that can accurately simulate drivers’ car-following behaviors is helpful to the development of driving assistance systems and autonomous driving. Recent studies have shown the advantages of applying reinforcement learning methods in car-following modeling. However, a problem has remained where it is difficult to manually determine the reward function. This paper proposes a novel car-following model based on generative adversarial imitation learning. The proposed model can learn the strategy from drivers’ demonstrations without specifying the reward. Gated recurrent units was incorporated in the actor-critic network to enable the model to use historical information. Drivers’ car-following data collected by a test vehicle equipped with a millimeter-wave radar and controller area network acquisition card was used. The participants were divided into two driving styles by K-means with time-headway and time-headway when braking used as input features. Adopting five-fold cross-validation for model evaluation, the results show that the proposed model can reproduce drivers’ car-following trajectories and driving styles more accurately than the intelligent driver model and the recurrent neural network-based model, with the lowest average spacing error (19.40%) and speed validation error (5.57%), as well as the lowest Kullback-Leibler divergences of the two indicators used for driving style clustering.

## 1. Introduction

With the process of urbanization and the rapid growth in the number of vehicles, car following has become the most common driving behavior in daily driving. It is of great significance to build a model that can accurately simulate drivers’ car-following behaviors. Car-following models have been widely applied in microscopic traffic simulation, the tests of driving assistant systems, and other fields [1,2,3]. In the future of autonomous driving, designing an autonomous vehicle that can achieve human-like driving can also improve the riding comfort and trust of passengers [4,5].

Research on car-following modeling has lasted for more than half a century. The modeling methods can be divided into two types: theoretical-driven methods and data-driven methods [6]. The former generally puts forward various hypotheses on drivers’ reactions based on the observation of actual car-following behaviors and establishes mathematical relationships to express drivers’ car-following strategies. In the expression of theoretical-driven models, there are some parameters representing the preferences of different drivers. Since the theoretical-driven models contain rather few parameters, their expression ability is limited, thus the accuracy and generalization ability cannot be ensured [7]. Therefore, in recent years, a large number of studies have used data-driven methods to build car-following models [7,8,9,10].

The data-driven methods can directly use data to learn drivers’ control strategies and have fewer assumptions on driving behavior. Data-driven models such as neural networks have stronger expression ability and can be easily updated. Studies have shown that the data-driven car-following models have better generalization ability than the theoretical-driven models [7,11]. In the related research of data-driven models, directly applying learning methods to learn the map from states to actions is referred to as behavior cloning (BC) [12]. Although the BC models have been proven to be effective in some studies, they may suffer from the problem of cascading errors, which is very common in sequential decision-making problems [13]. The error existing in the single time-step prediction will accumulate and gradually increase in the sequential decision-making process, which may cause the model to reach unseen states, making the model have even worse predictions. To avoid this problem, some researchers have begun to use reinforcement learning (RL) methods [11,14].

RL assumes that human behavior maximizes the cumulative long-term rewards [15]. When the reward is determined, the strategy learned by using RL algorithms can have the ability of long-term planning and can avoid cascading errors [16]. However, the designing of the reward function in RL is very hard, manually designing it is time-consuming, and an inappropriate reward function may even lead to unexpected behaviors [17,18]. To overcome this issue, inverse reinforcement learning (IRL) has been proposed [19]. Instead of modeling the strategy directly, the IRL method first learns the reward function and then obtains the strategy by using RL.

The traditional IRL method often uses a linear representation of reward function, which may not reflect drivers’ nonlinear intrinsic preferences [20]. Besides, it requires a lot of computation power as it needs to solve the RL subprocess during training [21]. In this study, a recently proposed algorithm called generative adversarial imitation learning (GAIL) was applied to model drivers’ car-following strategies. The main contributions of this study are as follows: (1) a novel way to model drivers’ car-following behaviors was proposed. The proposed model uses a nonlinear function that uses neural networks to automatically learn drivers’ rewards and strategies, and the training of the model does not need to solve the RL subprocess, which can save a lot of computation power. Besides, a kind of recurrent neural network (RNN) called gated recurrent units (GRU) is used in the proposed model to fit drivers’ car-following policy, which is able to take advantage of the historical information for time-sequence prediction. (2) The car-following data of drivers collected in real-world situations is used to train and verify the proposed model and two other representative car-following models. Different methods were compared in terms of replicating drivers’ car-following trajectories and driving styles.

The rest of the paper is organized as follows: Section 2 briefly reviews the related work and backgrounds of this study. Section 3 presents the architecture of the proposed model and the algorithm. Section 4 describes the experiments and data used in this study. Section 5 presents the details of model training and validation. Section 6 presents the results and comparisons of different methods. The final section presents the discussion.

## 2. Related Work

### 2.1. Theoretical-Driven Car-Following Models

The development of theoretical-driven car-following models can be dated back to the 1960s. Pipes [22] and Chandler, et al. [23] proposed the earlier General Motors models. Since then, several different models have been proposed, such as the Gipps model [24], the optimal velocity model [25], and the intelligent driver model (IDM [26]). The detailed development of theoretical-driven car-following models can be found in [27,28].

Among the mentioned models, IDM is one of the most widely used car-following models, which has the formulations as presented in Equations (1) and (2). In this paper, IDM was calibrated and compared with the proposed model.
(1)$$a_{t}=a_{\max}\left[1-{\left(\frac{v_{t}}{v_{f}}\right)}^{\beta }-{\left(\frac{h_{t}^{des}}{h_{t}}\right)}^{\beta }\right]$$
(2)$$h_{t}^{des}=h_{jam}+\max(0,v_{t}\cdot TH_{des}-\frac{v_{t}\cdot \Delta v_{t}}{2\cdot \sqrt{a_{\max}\cdot a_{com}}})$$
where $a_{\max}$ and $a_{com}$ are the maximum desired acceleration and comfortable deceleration, $v_{f}$ is the maximum desired speed, $TH_{des}$ is the desired time headway, $h_{jam}$ is the minimum jam spacing, and *β* is the acceleration parameter. $v_{t}$, $h_{t}$, $h_{t}^{des}$, $\Delta v_{t}$, and $a_{t}$ refer to the speed, spacing, desired spacing, relative speed, and acceleration at time *t*. 

### 2.2. Behavior Cloning Car-Following Models

The earlier study using behavior cloning (BC) methods to model drivers’ car-following behaviors was conducted by Jia et al. [29]. The authors of the study proposed a neural network with two hidden layers that took inputs including speed, relative speed, spacing, and desired speed, and gave the acceleration as the output. Chong et al. [30] proposed a similar BC model but with only one hidden layer. The type of neural networks adopted in these studies is often called fully connected neural networks (FCN), which normally takes only a single time-step of relevant information as the input. For a sequential decision modeling problem like car-following, RNNs, which can use historical information, have gained popularity in recent years. For example, in the study conducted by Zhou et al. [10], an RNN model was proposed that was found to have better adaptability in oscillating traffic circumstances compared to the FCN approaches. Li et al. [6] applied a more advanced type of RNN called long short-term memory (LSTM) for modeling car-following. GRUs, which is a simpler version of LSTM, were adopted in the study conducted by Wang et al. [7] for modeling drivers’ car-following behaviors. Except for neural networks, there are studies applying other learning methods like Gaussian mixture models and hidden Markov models for modeling car-following behavior [31,32]. In this study, the RNN model proposed by Zhou et al. [10] was selected as a comparative method.

### 2.3. Reinforcement Learning

In the setting of RL, the Markov-decision process (MDP) is used to model a sequential decision-making problem. An MDP can be defined as a tuple {*S, A, T, R, γ*}, where *S* denotes the state space, *A* denotes the action space, *T* denotes the transition matrix, *R* denotes the reward function, and *γ* denotes the discount factor. RL assumes an agent exists in the predefined environment. At every time-step, the agent observes a state and takes an action following its policy, gets to the next state according to the transition matrix, and it then receives a reward.

To solve an MDP using RL, the reward function must be specified, then the policy can be obtained by applying value-based or policy-based algorithms [15]. As for the study applying RL for modeling car-following, Zhu et al. [11] adopted a deep deterministic policy gradient model in which the reward function was determined as speed discrepancies between the simulated trajectories and the test data.

In this study, the MDP for modeling car-following is defined as follows: at a certain time-step *t*, the state includes the speed of the following vehicle denoted as $v_{t}$, the spacing between the follower and the leading vehicle denoted as $h_{t}$ and the relative speed between the follower and the leader denoted as $\Delta v_{t}$. The action is the longitudinal acceleration of the follower denoted as $a_{t}$. A kinematic model is adopted as the state transition matrix (Equation (3)). The reward does not need to be defined in this study because the proposed model can learn it from the data. The discount factor γ was set as 0.99 in this study.
(3)$$\begin{array}{l} v(t+1)=v(t)+a(t)\cdot dt\\ \Delta v(t+1)=v_{lead}(t+1)-v(t+1)\\ h(t+1)=h(t)+\frac{\Delta v(t)+\Delta v(t+1)}{2}\cdot dt \end{array}$$
where d*t* is the simulation time interval which was set as 0.1 s in the present study, and $v_{lead}$ denotes the speed of the leader that was externally inputted from the collected data.

### 2.4. Inverse Reinforcement Learning

The goal of IRL is to learn the reward function given a set of expert demonstrations. Many approaches have been proposed to solve this problem [33]. The framework of maximum entropy IRL (Max-Ent IRL) proposed in [34] has been accepted by many studies in recent years. In the formulation of Max-Ent IRL, the probability of a trajectory is proportional to the sum of the exponential rewards gained along the trajectory (Equation (4)). Therefore, trajectories with higher rewards become exponentially more likely to happen, while trajectories with lower rewards can still happen but with exponentially fewer probabilities. As proved in [35], the principle of maximum entropy can handle the suboptimality of human behavior.
(4)$$p\left(\tau \right)∼\exp\left(\sum_{s,a\in \tau }r\left(s,a\right)\right)$$ where $p\left(\tau \right)$ denotes the probability of a certain trajectory, and $r\left(s,a\right)$ denotes the return of state-action pairs.

With the above formulation of trajectory distribution, the reward function represented by a linear representation or by a neural network can be learned by maximizing the log-likelihood of demonstrators’ trajectories. However, the Max-Ent-IRL typically requires expensive computational power as it needs to solve the forward problem of finding an optimal policy with respect to the current reward [21], making it difficult to apply to MDPs with high dimensional state space.

### 2.5. Generative Adversarial Imitation Learning

Since the basic ideas of IRL and generative adversarial nets (GANs) proposed by Goodfellow et al. [36] have a lot in common [37], generative adversarial imitation learning (GAIL) combines IRL with GANs, and extends Max-Ent-IRL by using the framework of GANs [38]. GAIL retains the advantages of Max-Ent-IRL. At the same time, GAIL uses a surrogate reward and directly imitates the demonstrator’s behavior through policy optimization. GAIL has the advantage of low computation cost and can be better applied to complex problems. In the study conducted by Kuefler et al. [16], GAIL was applied to model drivers’ behavior in highway driving using the next-generation simulation (NGSIM) datasets. Comparing with this study, the present study used a different architecture of GAIL, and different driving styles were modeled using the datasets after clustering.

## 3. The Proposed Model

The structure of the proposed model denoted as GAIL-GRU [16,39] is shown in Figure 1 which is composed of two parts: (1) a generator that consists of an actor-critic structure and the car-following environment, which generates the faked state-action pairs, and (2) a discriminator that functions as classifying the generated state-action pairs and the actual state-action pairs obtained from the collected data. Also, the discriminator outputs the reward signal for the car-following environment. In this section, the two components of the model and the algorithm are introduced respectively.

### 3.1. The Generator

An actor-critic structure, which is the core component of the generator, was adopted as an agent to interact with the car-following environment. As can be seen in Figure 2, the actor in the upper part and critic in the lower part share the same feature extraction layer for reducing the parameters of the model. As mentioned in Section 2, the RNN that can use historical information has been widely applied in car-following modeling. Therefore, this study adopted a GRU in the feature extraction layer in which the number of neurons was set to 60. After the feature extraction layer, FCN was used to convert the output of GRU to a single value. Exponential linear units (ELUs) are used as the activation function between each layer as ELUs have been proven to overcome the vanishing gradient problem [40]. The inputs of the actor-critic is a time series state ${\left\{s_{t-N},s_{t-N+1},\ldots ,s_{t}\right\}}_{t}$ in which the sequence length N was set to 10. The actor outputs the policy of the agent which is represented by a normal distribution$N\left(\mu_{t},\sigma \right)$, and the critic outputs the value $V_{t}$ which represents the expected long-term reward of the input state.

The proximal policy optimization (PPO) algorithm proposed in [41] was used to update the parameters of the actor-critic networks (Equations (5)–(9)). The PPO algorithm is a policy-based RL algorithm, which uses actor-critic as its basic architecture. The PPO algorithm can effectively solve the problems of high variance and difficult convergence in the training of a policy-based RL algorithm by limiting the range of policy updates in every single step, avoiding the excessive change of policy, and making the policy improve steadily.
(5)$$L\left(\theta \right)=\hat{E_{t}}\left[L_{t}^{CLIP}\left(\theta \right)-c_{1}L_{t}^{VF}\left(\theta \right)+c_{2}E[\pi_{\theta }](s_{t})\right]$$
(6)$$L_{t}^{CLIP}\left(\theta \right)=\hat{E_{t}}\left[\min(\frac{\pi_{\theta }\left(a_{t}|s_{t}\right)}{\pi_{\theta_{old}}\left(a_{t}|s_{t}\right)}\hat{A_{t}},clip(\frac{\pi_{\theta }\left(a_{t}|s_{t}\right)}{\pi_{\theta_{old}}\left(a_{t}|s_{t}\right)},1-\varepsilon ,1+\varepsilon )\hat{A_{t}}\right]$$
(7)$$L_{t}^{VF}\left(\theta \right)={\left(\sum_{t^{’}>t}\gamma^{t’-t}r_{t^{’}}-V_{\theta }\left(s_{t}\right)\right)}^{2}$$
(8)$$clip(x,a,b)=\left\{\begin{array}{l} a\;\;if\;x\leq a\\ x\;\;if\;a<x<b\\ b\;\;if\;x\geq b \end{array}\right.$$
(9)$${\hat{A}}_{t}=\sum_{t^{’}>t}\gamma^{t’-t}r_{t’}-V_{\theta }\left(s_{t}\right)$$
where $L\left(\theta \right)$ is the loss function of the actor-critic networks; $L_{t}^{CLIP}\left(\theta \right)$ denotes the loss for the actor and $L_{t}^{VF}\left(\theta \right)$ denotes the loss for the critic; $V_{\theta }\left(s_{t}\right)$ denotes the value of state $s_{t}$; ${\hat{A}}_{t}$ denotes the advantage value of state $s_{t}$; θ is the parameters of the actor-critic networks; ${\hat{E}}_{t}$ denotes the expectation operation; $c_{1},c_{2},\varepsilon$ are all parameters which all set to 0.5, 0.01, and 0.2, respectively; $\pi_{\theta_{old}}\left(a_{t}|s_{t}\right)$ is the last output of the actor; and $E[\pi_{\theta }](s_{t})$ denotes the entropy of policy $\pi_{\theta }$. 

### 3.2. The Discriminator

The discriminator is an FCN with two hidden layers as depicted in Figure 3. Each hidden layer has 64 neurons. ELUs were used as the activation function except for the last layer, which used the sigmoid function. The input of the discriminator is state-action pairs, and the output is the probability of the input belonging to each category. The generated state-action pairs and the actual state-action pairs were marked as 1 and 0, respectively. The cross-entropy loss function commonly used in the binary classification problem, as shown in Equation (10), was used to update the parameters of the discriminator.
(10)$$L(\omega )=\frac{1}{N}\sum_{i}-\left[y^{i}\cdot \log\left(D_{\omega }\left(s^{i},a^{i}\right)\right)+(1-y^{i})\cdot \log\left(1-D_{\omega }\left(s^{i},a^{i}\right)\right)\right]$$
where L(ω) is the loss of the discriminator networks, N is the number of the training samples, $y^{i}$ is the label of sample i, $\left(s^{i},a^{i}\right)$ is the *i*th state-action pair, and $D_{\omega }\left(s^{i},a^{i}\right)$ denotes the output of the discriminator parameterized by ω.

The discriminator also provides the reward signals for the car-following environment. The state-action reward $r(s_{t},a_{t})$ that denotes the return of taking action $a_{t}$ in the state $s_{t}$ is determined by Equation (11). It can be inferred that when the generated state-action samples get closer to the actual samples, the output of the discriminator for the generated samples becomes closer to 0, and the reward becomes higher. Therefore, the reward signal provided by the discriminator is actually encouraging the agent to act more similarly to the demonstrator.
(11)$$r(s_{t},a_{t})=-\log(D_{\omega }(s_{t},a_{t}))$$

### 3.3. The Proposed Algorithm

The proposed algorithm is showed presented in Algorithm 1 below. The machine learning package PyTorch (Facebook, Menlo Park, CA, USA) was used for implementing the proposed model. The Adam optimizer [42] was used for training the model. In every training episode, the discriminator was trained for more times than that of the actor-critic networks to facilitate the convergence of the entire model [43].
**Algorithm 1:** Algorithm for modeling drivers’ car-following behaviors**Input:** The collected state-action pairs of drivers denoted as *D* Output: 1: The strategy of drivers denoted as π
 2: **Algorithm begins:** 3: Randomly initialize the parameters of the discriminator and the actor-critic networks as $\theta_{0}$ and $\omega_{0}$. 4: For i=0 to *N*, repeat the following steps 5: Using the policy $\pi_{i}$ output by the actor-critic parameterized by $\omega_{i}$ to interact with the car-following environment, record the interaction history as the generated state-action pairs $D_{G_{i}}$ 6: Update the parameters of the discriminator for three times from $\theta_{i}$ to $\theta_{i+1}$ with the generated samples $D_{G_{i}}$ and the actual samples D using the Adam rule and the loss L(ω). 7: Use the updated discriminator parameterized by $\theta_{i+1}$ to provide rewards (Equation (11)) for the environment, update the parameters of the actor-critic networks from $\omega_{i}$ to $\omega_{i+1}$ using the Adam rule and the loss $L\left(\theta \right)$. 8: **Algorithm end**

## 4. Data Description

### 4.1. The Experiments

Data from two field tests conducted in Huzhou City in Zhejiang Province and Xi’an city in Shaanxi Province in China were used in this study. The driving route included urban roads and expressways, as shown in Figure 4. The urban section is part of the Huzhou section of national highway 104 (where the speed limit is 70 km/h) and part of the east third ring road and south third ring road of Xi’an (where the speed limit is 100 km/h), and the highway section is the Huzhou-Changxing section of the G25 highway (where the speed limit is 120 km/h) and the Xiangwang-Lantian section of G40 highway (where the speed limit is 120 km/h).

A total of 42 drivers participated in the experiment. All of the subjects were male, the average age was 40.3 years old and the standard deviation was 5.1. The participants’ driving experiences ranged from 2 to 30 years with the average being 15.2 years and the standard deviation was 7.5. During the test, all of the subjects were asked to drive the test vehicle according to their personal driving styles, and they were only told the starting and ending positions. It took 90–110 min for the subjects to complete the entire test.

The test data were collected by a test vehicle equipped with a variety of sensors as exhibited in Figure 5. Among them, the millimeter-wave radar can collect the motion states of the surrounding vehicles during driving including the spacing and relative speed with the targets. The detection angle is ±10° and 64 targets around the test vehicle can be tracked at the same time. The speed of the test vehicle was collected from the CAN acquisition card. All of the recorded data were sampled at 10 Hz. During the tests, the real-time pictures of drivers’ faces, as well as the front, back, and sides of the vehicle, were captured by the video monitoring system of the vehicle, which facilities the verification of uncertain car-following periods.

### 4.2. Car-Following Periods Extraction

In this study, the rules for determining a car-following period implemented in a previous study were adopted [44]. The details include the following: (1) the test vehicle should follow the same leading car throughout the entire period; (2) the distance between the test vehicle and leading vehicle should be less than 120 m; (3) the test vehicle and leading vehicle should be in the same lane; and (4) the length of the car-following periods should be greater than 15 s. A program was designed to filter the data segments that meet the above criteria, and then the filtered segments were checked by manually playing back the video recorded by the front camera of the test vehicle. The checking procedure includes the confirmation of the validation of every period and the determination of the starting point and the endpoint of each car-following period. Finally, a total of nearly one thousand car-following periods were extracted from the collected data. The extracted car-following data included the speed of the ego vehicle, the spacing from the leading vehicle, and the relative speed between the ego vehicle and leading vehicle which were all measured by the instrumented vehicle.

Two steps were taken to process the extracted data. First, all of the data were processed by a moving average filter to eliminate the noise in the data [45], where the length of the sliding window was set as 1 s. Second, because acceleration is also needed to train the model, this study adopted a Kalman filter for estimating acceleration because simply differentiating the speed cannot achieve an accurate estimation of the acceleration [46]. Figure 6 shows the speed in one car-following period randomly selected from the dataset. Figure 7 depicts the results of the acceleration estimated from the raw signal using the two different approaches mentioned above. It can be observed that the estimation results of the Kalman filter in blue is smoother and more accurate than that of differentiating the speed in red.

### 4.3. Driving Style Clustering Based on K-Means

Because the proposed model is based on RL, it assumes that the agent that should be modeled has certain kinds of preferences or rewards. Every individual driver may have different preferences during car-following. However, in the literature of driving-behavior research, it is common to cluster drivers into different driving styles such as aggressive or conservative. Drivers belonging to one group may have similar preferences. So, in this study, instead of using the car-following data of each participant to train the proposed model, K-means was initially applied to cluster the participants into different driving styles. The data from the different groups were then used for model training.

There are many indicators that can be used to characterize the driving style of drivers in car-following. Except for basic kinematic indicators, such as spacing, speed, or relative speed, time-based variables such as time headway (TH) and time-to-collision (TTC) have also been accepted for characterizing driving styles [47,48]. In the study conducted by Liu et al. [49], mean TH and mean TH when braking were adopted as features to classify drivers into different driving styles. In the present study, different combinations of the above-mentioned features were selected as the inputs for the K-means algorithm, the silhouette coefficient was used to evaluate the quality of the clustering results [50]. Before carrying out clustering, the car-following dataset was first normalized to eliminate the influence of dimension. Finally, mean TH and mean TH when braking were determined as feature vectors for clustering as the combination achieved the highest silhouette coefficient among all other choices. According to the value of the silhouette coefficient, the number of clusters was also determined to be two. Figure 8 and Figure 9 present the comparison of mean TH and mean TH when braking of the two groups of drivers after clustering. T-tests showed that the conservative group, which consisted of 16 drivers, maintained a significantly higher mean TH (*t* = 6.748, *p* < 0.001) and mean TH when braking (*t* = 7.655, *p* < 0.001) than the aggressive group, which consisted of 26 drivers.

Figure 10 shows the changes for mean acceleration for the two groups of drivers in different ranges of relative speed during car-following. The aggressive drivers kept higher acceleration compared to the conservative drivers in all the ranges except in the range of 2–4 m/s. Table 1 presents the descriptive statistics of car-following data from the two groups of drivers. The conservative drivers maintained a lower mean speed and acceleration, with a larger mean spacing compared to the aggressive drivers. It can be seen that the two groups of drivers exhibited distinct driving styles.

## 5. Model Training and Evaluation

### 5.1. Simulation Setup

The initial state $s_{0}$ which included the speed, spacing, and relative speed of the following vehicle, was externally input from the empirical data to enable the RL agent to interact with the car-following environment. Following this, the RL agent chooses an action based on its policy $a_{t}∼\pi (a_{t}|s_{t})$, and the future states of the following vehicle were updated by the state-transition matrix $s_{t+1}=T(s_{t+1}|a_{t},s_{t})$. The simulation for every car-following period terminated when it reached its maximum time-step.

### 5.2. Evaluation Matrix

#### 5.2.1. Root Mean Square Percentage Error

As suggested in [51], this study used the root mean square percentage error (RMSPE) of spacing and speed between simulation results and actual data to evaluate the performance of car-following models. The evaluation matrices are as follows:(12)$$RMSPE(v)=\sqrt{\frac{\sum_{i=1}^{N}{\left(v_{i}^{sim}-v_{i}^{obs}\right)}^{2}}{\sum_{i=1}^{N}{\left(v_{i}^{obs}\right)}^{2}}}$$
(13)$$RMSPE(h)=\sqrt{\frac{\sum_{i=1}^{N}{\left(h_{i}^{sim}-h_{i}^{obs}\right)}^{2}}{\sum_{i=1}^{N}{\left(h_{i}^{obs}\right)}^{2}}}$$ where RMSPE(v) and RMSPE(h) denote the RMSPE of speed and spacing, respectively; $v_{i}^{obs},h_{i}^{obs}$ denote the empirical speed and spacing in *i*th time; and $v_{i}^{sim},h_{i}^{sim}$ denote the simulated speed and spacing in *i*th time. 

#### 5.2.2. Kullback-Leibler Divergence

Kullback-Leibler (KL) divergence is a measure of the difference between two distributions that can be calculated by Equation (14). When the value of the KL divergence is smaller, the two distributions become more similar. The car-following models should produce a similar distribution of representative features with the drivers’ demonstrations. Because this study used mean TH and mean TH when braking to divide the drivers into two different driving styles, the similarity between the distribution of the simulated mean TH and mean TH when braking and the empirical distributions were measured by calculating the KL divergence to evaluate the performance of car-following models in reproducing drivers’ driving styles.
(14)$$KL(P,Q)=\sum P(x)\log\frac{P(x)}{Q(x)}$$
where P,Q denotes two distributions.

### 5.3. Cross-Validation

K-Fold cross-validation was adopted in this study to evaluate the generalization ability of car-following models. Specifically, *k* was determined to be 5 in this study due to the scale of the dataset. The dataset of aggressive and conservative drivers was equally divided into 5 groups. As illustrated in Figure 11, every fold was taken as the test set, and the remaining four folds were taken as the training set. The procedure was repeated five times, and then the average performance of RMSPE was calculated for the training sets and the test sets.

### 5.4. Models Investigated

As mentioned in Section 2, the representative theoretical-driven model IDM and recent behavior cloning model RNN were chosen to be compared with the proposed model.

#### 5.4.1. IDM

The parameters of IDM were calibrated with the objective of minimizing the RMSPE of spacing as Equation (15). A large penalty of the crash was also added in the objective function, as no collision was observed in the collected data. In this paper, the population size and the maximum iteration times of the genetic algorithm (GA) are set to 100. In order to reduce the influence of the randomness of GA on the calibration of IDM, the GA algorithm is repeated 12 times, and the final result is the combination of parameters with the minimum error.
(15)$$c(IDM)=RMSPE(h)+\lambda \cdot n_{crash}$$
where RMSPE(h) denotes the RMSPE of spacing, $n_{crash}$ denotes the number of crashes when applying the calibrated IDM model for simulation, and λ is the parameter for the crash penalty.

#### 5.4.2. RNN Based Model

In accordance with another study [10], an RNN with 60 neurons for its hidden layer was built. The model takes a time sequence of 1 s as the input. The gradient descent algorithm was used to train the model with the cost function defined as follows.
(16)c(RNN)=RMSPE(v)+RMSPE(h)
where RMSPE(v) and RMSPE(h) denotes the RMSPE of speed and spacing respectively.

## 6. Results

The performances of the proposed model GAIL-GRU, the theoretical-driven model IDM, and the recent behavior-cloning model RNN for replicating the car-following trajectories of the drivers with different driving styles on the training sets and test sets are presented in Figure 12 and Figure 13, respectively. As can be observed, GAIL-GRU outperformed the other two models as it achieved the lowest training error and test error for speed and spacing in both datasets. The RNN model has the highest training error and test error among the three models. For IDM, even though it exhibited a similar performance with GAIL-GRU on the training sets, it failed to generalize to the same level as GAIL-GRU on the test sets.

The three models trained by the same data from the aggressive dataset were used for simulation to illustrate and compare the characters of the three models in reproducing drivers’ car-following trajectories. The simulation results of spacing for three car-following periods randomly selected from the dataset are presented in Figure 14. As can be seen, the proposed GAIL-GRU tracks the empirical spacing more closely than the IDM and RNN methods. The simulation results of RNN and GAIL-GRU, which all make use of historical information, are similar in the first 10 s or so, but the deviation from the empirical data becomes increasingly larger for RNN as the simulation continues.

Figure 15 and Figure 16 present the comparison of the KL divergence for mean TH and mean TH when braking, respectively. As can be seen, the GAIL-GRU model can produce the closest distribution to the empirical data of the two indicators which were used to cluster the drivers into two different driving styles.

## 7. Discussion

The present study demonstrated that the proposed GAIL-GRU model can replicate human-like car-following behaviors and driving styles more accurately than the theoretical-driven model IDM and the behavior-cloning model RNN. Compared with IDM, the proposed model used the actor neural network which consists of two hidden layers including a GRU network and a fully connected network, to fit drivers’ car-following policies, while IDM only contains six parameters to be calibrated. Significantly more parameters were contained in the proposed model, accounting for a better fitting ability and generalization ability compared to the IDM.

Compared with RNN, the proposed model also used a kind of RNN to represent drivers’ strategies, and the length of the input was also set to be 1 s. However, the RNN model has a higher error for replicating car-following trajectories. From the simulation results of spacing for three car-following periods, it can be seen that the RNN experienced the problem of cascading errors, while the proposed model did not have this problem. The reason for this is that the RNN only learns the state-action relationships during training. When the input for RNN is not included in the training data, the output of RNN may deviate from drivers’ actual control actions. The proposed model is based on RL, the strategy output by the proposed model has the objective of maximizing the accumulative rewards, which award behavior similar to the demonstrator. Therefore, the strategy obtained by the proposed model can be extended to unseen states.

Comparing RNN with IDM, the error of the BC model is higher than the theoretical-driven model, which seems to be inconsistent with previous studies [7,11]. However, it must be noted that the RMSPE of RNN has large variances. In the five-fold cross-validation tests, RNN achieved a comparatively low error with IDM in some of the validation, while obtaining a much higher error in others. It seems that the performance of RNN is largely dependent on the division of training and test data. This issue has also been mentioned in the study conducted by Wang et al. [7], which pointed out that the selection of training data has a great impact on the performance of purely data-driven BC car-following models. The five-fold cross-validation methods adopted in this study can eliminate the influence of dataset division on the performance of car-following models, which can help to make a more reliable evaluation of the actual performance of the model in real applications.

The proposed model borrows the idea of GANs. In the fields of computer vision, GANs have been proven to be able to generate life-like images to the original image in recent years. In this study, the proposed model is demonstrated to be able to generate car-following strategies and driving styles closest to the drivers. It can be seen that the training process of the proposed model is very similar to that of GANs. The essence of the training process is the process of mutual confrontation between the generator and the discriminator. The training of the discriminator makes the discriminator distinguish the category of the samples more accurately. However, the training of the generator produces samples more similar to drivers’ actual behaviors, so as to cheat the discriminator. After training, the samples generated by the generator can be perfectly close to the distribution of the real samples, while the discriminator cannot correctly tell the difference between the generated samples from the actual samples. The proposed model can output the strategy that is identical to the drivers’ actual behaviors.

This study also had some important limitations. First, the car-following behaviors modeled in this study could only represent a small group of drivers. A broader sample is needed in future studies. Secondly, this study did not consider the influence of vehicles in adjacent lanes on drivers’ car-following behaviors. These factors will be fully considered in future research to obtain a more accurate description of drivers’ behaviors.

## Figures and Tables

**Figure 1 sensors-20-05034-f001:**
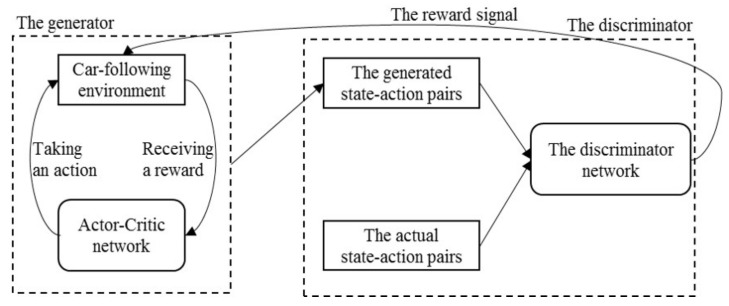
The structure of the proposed model generative adversarial imitation learning-gated recurrent unit (GAIL-GRU).

**Figure 2 sensors-20-05034-f002:**
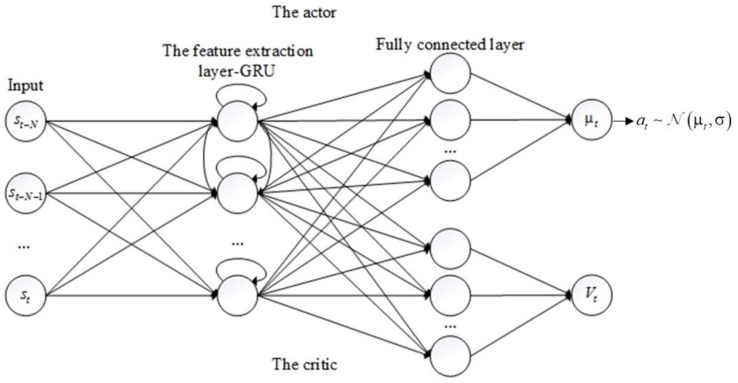
The architecture of the actor-critic structure.

**Figure 3 sensors-20-05034-f003:**
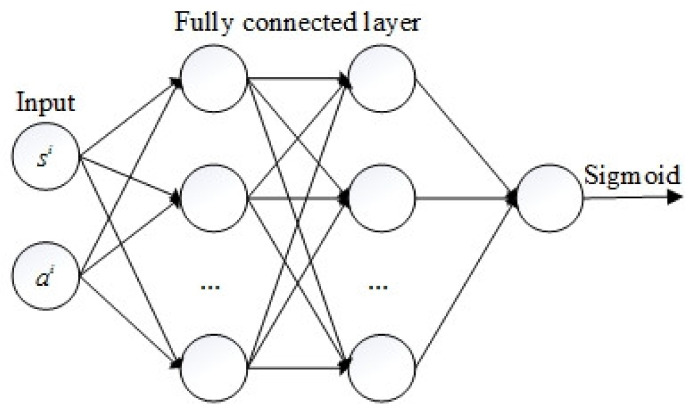
The architecture of the discriminator.

**Figure 4 sensors-20-05034-f004:**
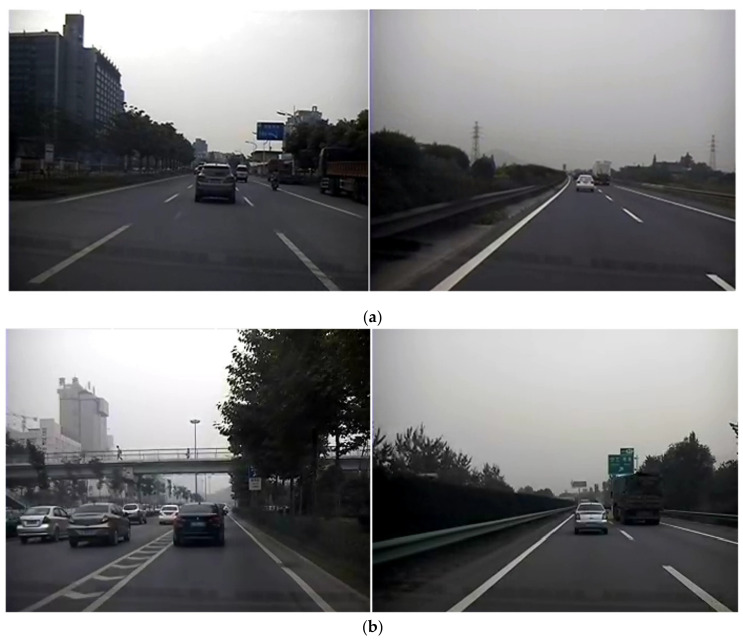
Driving scenarios in (**a**) Huzhou city and (**b**) Xi’an city.

**Figure 5 sensors-20-05034-f005:**
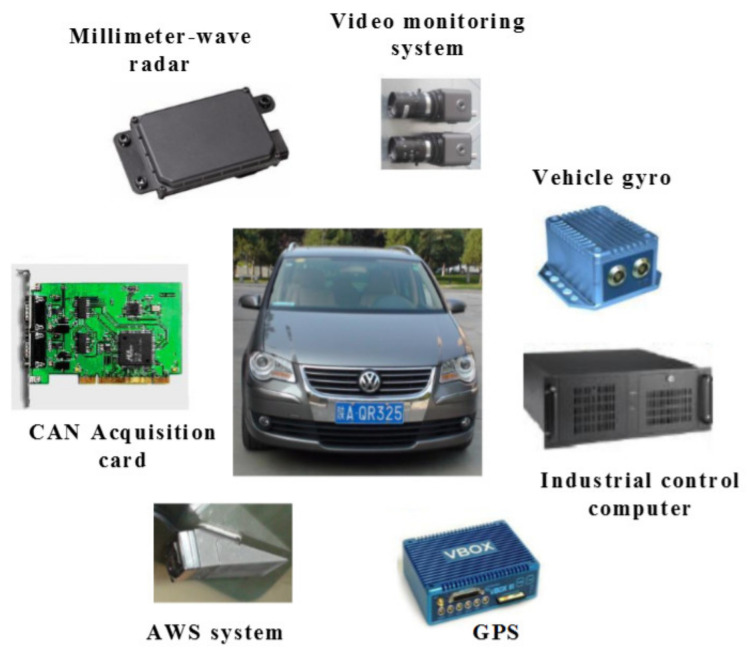
The test vehicle and related equipment.

**Figure 6 sensors-20-05034-f006:**
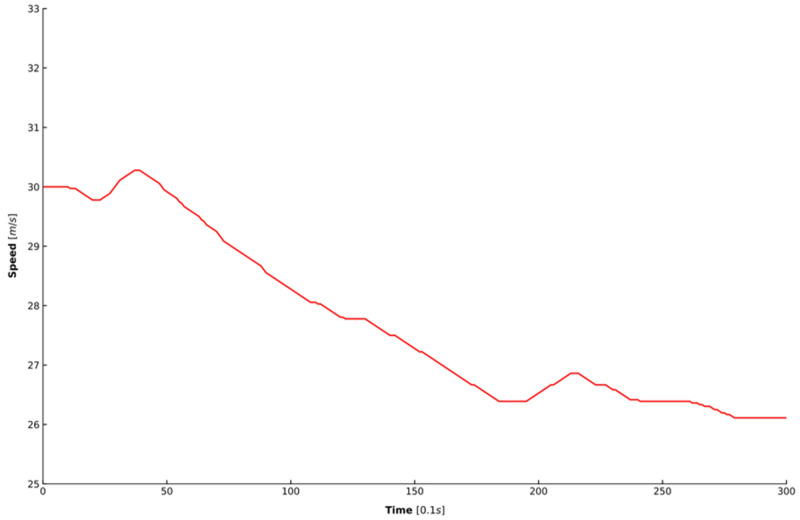
The speed in one car-following period.

**Figure 7 sensors-20-05034-f007:**
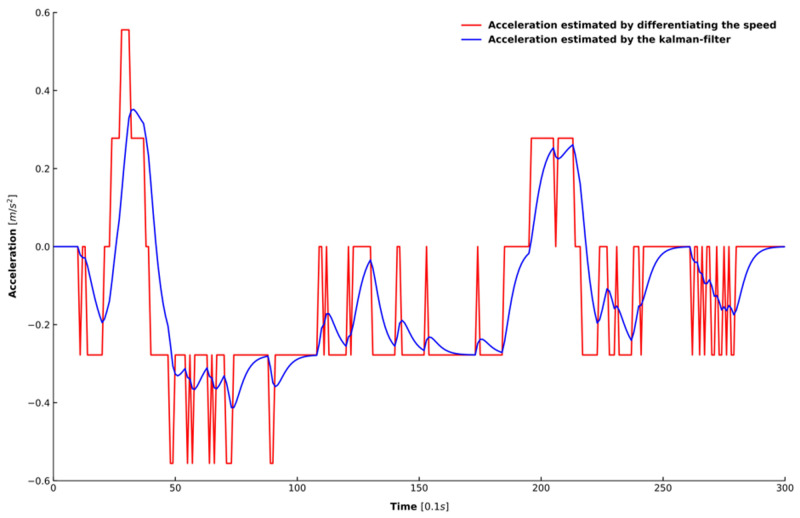
The comparison of different methods for estimating acceleration from the raw signal.

**Figure 8 sensors-20-05034-f008:**
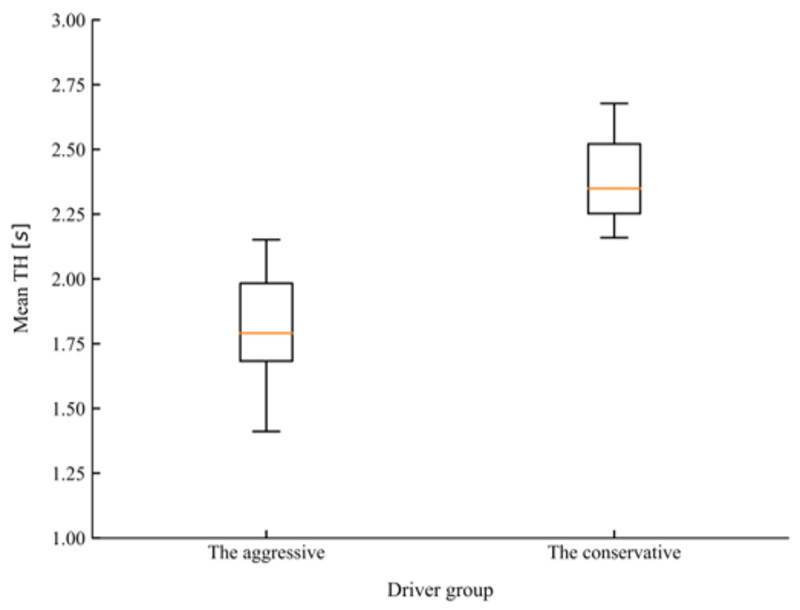
The comparison of mean time headway (TH) for the two driving styles.

**Figure 9 sensors-20-05034-f009:**
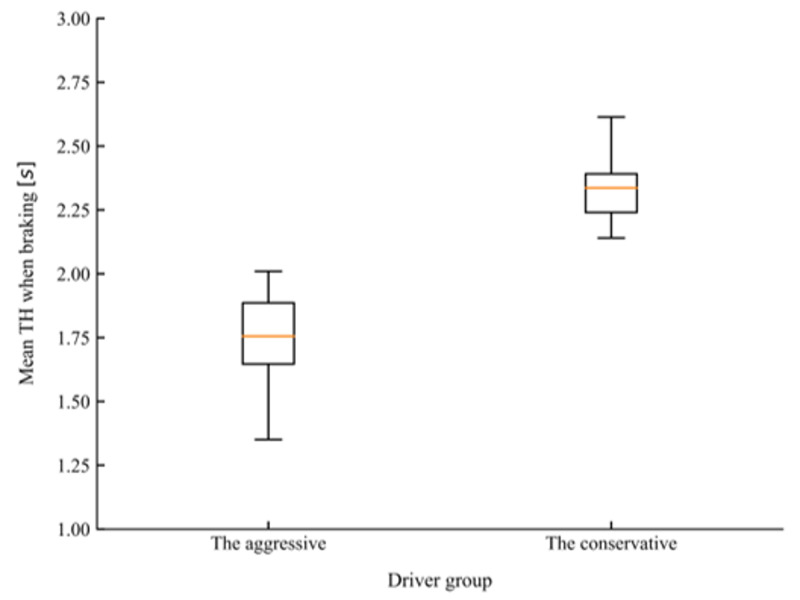
The comparison of mean time headway (TH) when braking for the two driving styles.

**Figure 10 sensors-20-05034-f010:**
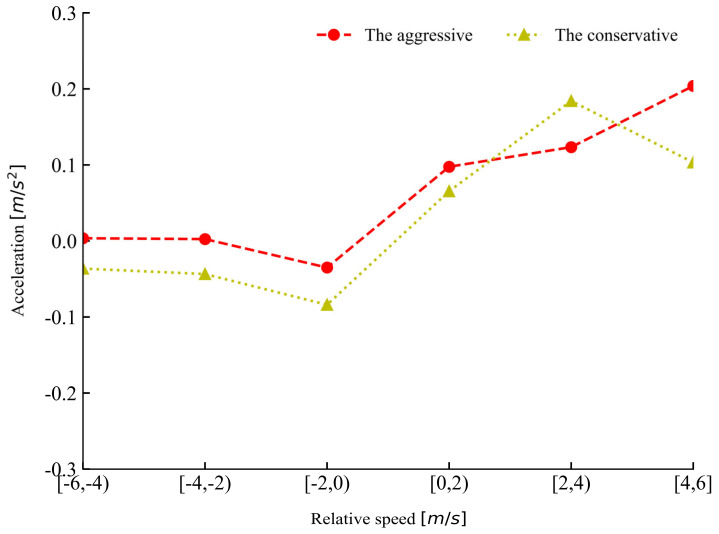
Rule of changes for acceleration in different relative speed ranges for the two groups of drivers.

**Figure 11 sensors-20-05034-f011:**
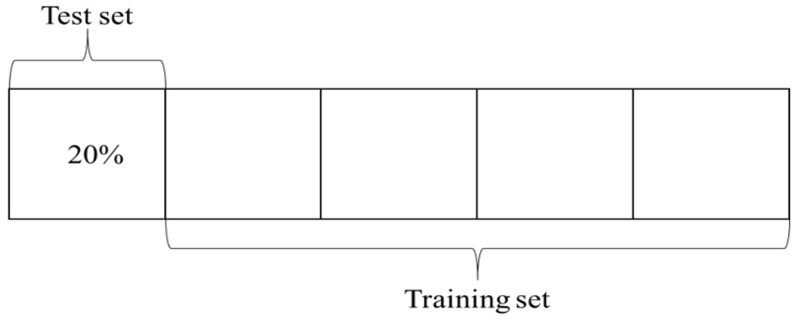
Five-fold cross-validation.

**Figure 12 sensors-20-05034-f012:**
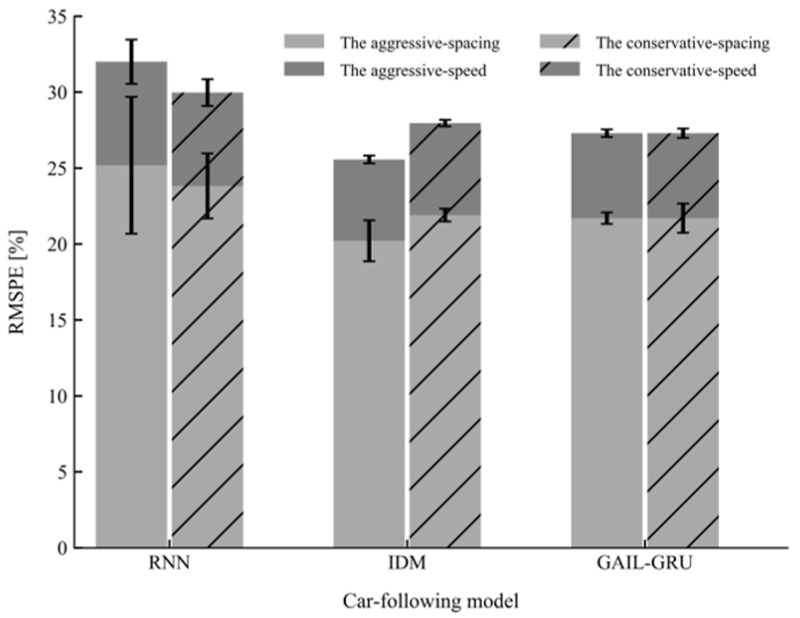
Mean and standard deviation of the root mean square percentage error (RMSPE) on the training sets for the three models.

**Figure 13 sensors-20-05034-f013:**
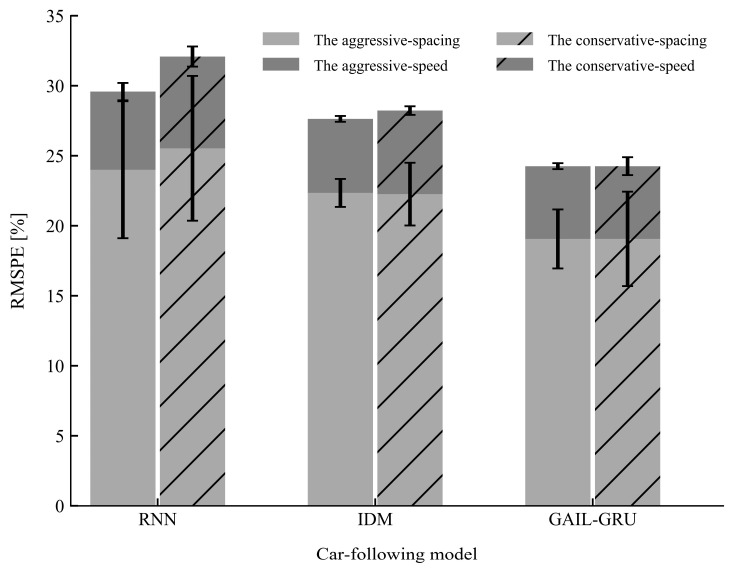
Mean and standard deviation of the root mean square percentage error (RMSPE) on the test sets for the three models.

**Figure 14 sensors-20-05034-f014:**
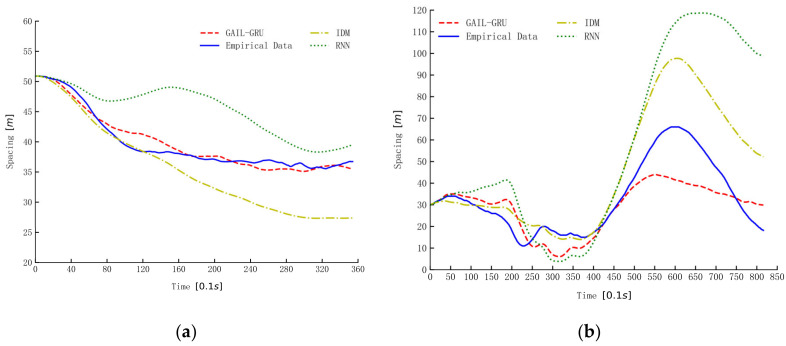
The simulation of spacing by different models. (**a**–**c**) are the simulation results of spacing for the three car-following periods randomly selected from the datasets.

**Figure 15 sensors-20-05034-f015:**
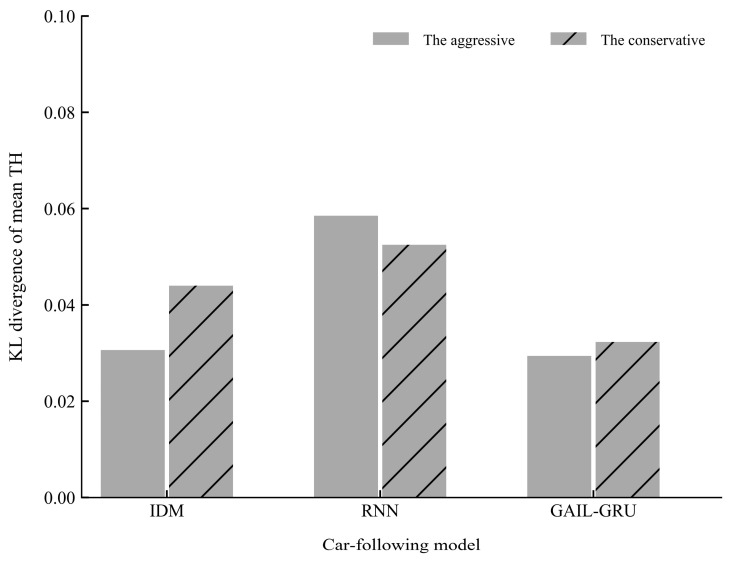
The Kullback-Leibler (KL) divergence of mean time headway (TH) for different models.

**Figure 16 sensors-20-05034-f016:**
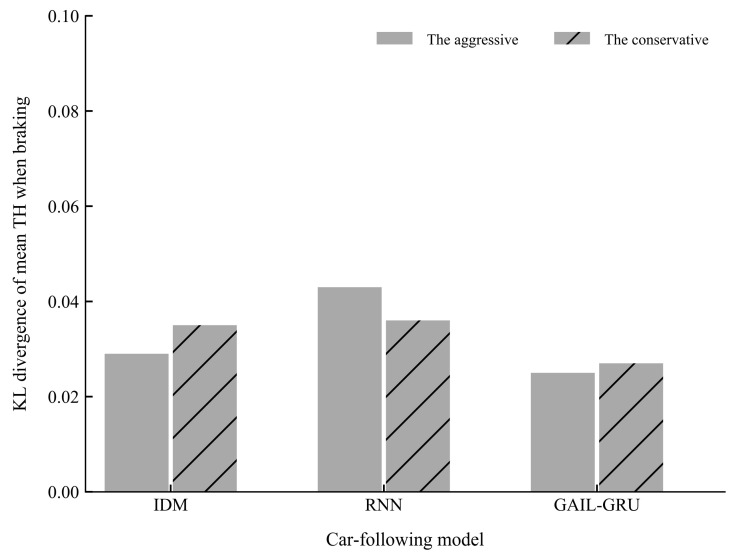
The Kullback-Leibler (KL) divergence of mean time headway (TH) when braking for different models.

**Table 1 sensors-20-05034-t001:** Descriptive statistics of the two groups of drivers in car-following.

Type	Spacing (m)			Speed (m/s)		
	Mean	Min	Max	Mean	Min	Max
Aggressive	34.15	2.27	120.00	18.57	2.50	33.70
Conservative	42.79	3.91	119.90	17.47	1.49	34.84
**Type**	**Acceleration (m/s^2^)**			**Relative Speed (m/s)**		
	**Mean**	**Min**	**Max**	**Mean**	**Min**	**Max**
Aggressive	−0.01	−3.14	1.64	−0.42	−12.45	5.80
Conservative	−0.04	−1.80	1.40	−0.52	−8.85	5.02

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
