# Peer review of "Modeling Car-Following Behaviors and Driving Styles with Generative Adversarial Imitation Learning"

_sensors, 2020, doi:10.3390/s20185034_

Round 1

Reviewer 1 Report

This paper proposes to use Generative Adversarial Imitation Learning (GAIL) to model car-following behavior. The topic and idea are interesting, and the rationale for employing GAIL for the problem is well justified. I think using GAIL for this problem is a very good fit.

The introduction is good. The background and motivation are also well described. Using GAIL for car following is explained well, and its advantages over rule-based models are adequately discussed.

However, in my opinion, the data that was used to evaluate the proposed method is not enough. The sample size is too small, with only 271 car-following instances. This is the biggest drawback of this paper. I think more data should be collected, or public car-following benchmark datasets can be utilized to validate the proposed method. 

If the following issues can be resolved, this paper will be a good contribution to the intelligent vehicles field.  

[Major issues]

  • Is RNN Gail novel? Is this setup/algorithm introduced the first time in this paper? Can we say that this is your novel contribution (line 71 page 2). How about the following paper [1] where they used GAIL and LSTM ( which is an RNN) . If this is not your novel contribution, then this statement should be rephrased or removed.

[1]Fernando, T., Denman, S., Sridharan, S., & Fookes, C. (2018, July). Learning Temporal Strategic Relationships using Generative Adversarial Imitation Learning. In Proceedings of the 17th International Conference on Autonomous Agents and MultiAgent Systems (pp. 113-121).

  • Why not use bigger, public datasets? Such as Us highway 101 dataset [2]? You only have 12 drivers. Alternatively, you can collect more data.

[2] Colyar, J., & Halkias, J. (2007). US highway 101 dataset. Federal Highway Administration (FHWA), Tech. Rep. FHWA-HRT-07-030.

  • Page 8, line 257, there are only 271 car-following cases. Isn't this number quite low for training a neural network? 
  • You used a simple RNN and IDM as your baselines. Can you also compare your method to a RL or IRL based driving behavior model? For example the following reference [3] uses a very similar approach to yours. What is your method's advantage over theirs? Can you at least discuss this point in your related works section?

[3]Kuefler, A., Morton, J., Wheeler, T., & Kochenderfer, M. (2017, June). Imitating driver behavior with generative adversarial networks. In 2017 IEEE Intelligent Vehicles Symposium (IV) (pp. 204-211). IEEE.

[Minor issues]

  • The resolution of the figures can be increased.
  • Page 3, line 128. Is the action discrete or continuous? Is it normalized between 0-1? 
  • Page 7, line 224. Can we safely say that the observed car following behavior is naturalistic? Drivers didn't drive in their natural driving setup, and they only drove the experimental car for a short amount of time. They were aware that they were observed. These factors may have affected their following behavior, and unnaturalistic patterns might have emerged. The number of drivers, 12, is also quite low for this kind of study. As an example, in SHRP2 project [4], more than 3400 drivers [4] driving behavior was collected. You don't have to do that much, but I would expect more than 12 different drivers to validate a learning-based model such as yours.

[4]PEREZ, M., MCLAUGHLIN, S., KONDO, T., ANTIN, J., MCCLAFFERTY, J., LEE, S., ... & DINGUS, T. (2016). Transportation safety meets Big Data: the SHRP 2 naturalistic driving database. Journal of the Society of Instrument and Control Engineers55(5), 415-421.

  • Figure 6 should also show the raw signal. We only see the processed output (if I am not mistaken.)
  • In my opinion, using K-means clustering for grouping drivers into aggressive/conservative with just 12 drivers is not convincing.

Reviewer 2 Report

This paper develops a learning-based car-following model by using generative adversarial imitation learning, which can considers the driving styles. Comparison experiments with real-world data are conducted to support their conclusions. In general, this paper has a good writing. Some comments are listed as follows:

  1. The related work in Section 2.2 are not sufficient. Behavior cloning approaches actually not only include the neural network, but it means directly copy the behavior from the data via some learning methods. For example, the hidden Markov models, Gaussian mixture regressions, etc. Therefore, to avoid misunderstanding, the state-of-the-art behavior cloning approaches in car-following scenarios should be discussed, for example, references with the following links:
    • https://ieeexplore.ieee.org/abstract/document/8879516
  2. Section 3, some symbols are lack of definition, for example, L(\theta), L_{t}^{CLP}, etc. Each symbol should be defined clear.
  3. In Algorithm 1, how to update the parameter theta from t to t+1?
  4. Also, how to update the w_{t}? More details should be given.

Round 2

Reviewer 1 Report

I am satisfied with the Authors' responses and edits. Good luck.